# Isogenic GAA-KO Murine Muscle Cell Lines Mimicking Severe Pompe Mutations as Preclinical Models for the Screening of Potential Gene Therapy Strategies

**DOI:** 10.3390/ijms23116298

**Published:** 2022-06-04

**Authors:** Araceli Aguilar-González, Juan Elías González-Correa, Eliana Barriocanal-Casado, Iris Ramos-Hernández, Miguel A. Lerma-Juárez, Sara Greco, Juan José Rodríguez-Sevilla, Francisco Javier Molina-Estévez, Valle Montalvo-Romeral, Giuseppe Ronzitti, Rosario María Sánchez-Martín, Francisco Martín, Pilar Muñoz

**Affiliations:** 1GENYO, Centre for Genomics and Oncological Research: Pfizer, University of Granada, Andalusian Regional Government PTS Granada-Avenida de la Ilustración 114, 18016 Granada, Spain; araceli.aguilar@genyo.es (A.A.-G.); juanegonzalezcorrea@gmail.com (J.E.G.-C.); eb3471@cumc.columbia.edu (E.B.-C.); iris.ramos@genyo.es (I.R.-H.); saragreco19@gmail.com (S.G.); jrodsevilla@gmail.com (J.J.R.-S.); javier.molina@genyo.es (F.J.M.-E.); rmsanchez@ugr.es (R.M.S.-M.); 2Department of Medicinal & Organic Chemistry and Excellence Research Unit of “Chemistry Applied to Biomedicine and the Environment”, Faculty of Pharmacy, University of Granada, Campus de Cartuja s/n, 18071 Granada, Spain; 3Instituto de Investigación del Hospital Universitario La Paz, IdiPAZ, 28029 Madrid, Spain; mlermajuarez@gmail.com; 4Fundación para la Investigación Biosanitaria de Andalucía Oriental-Alejandro Otero (FIBAO), 18012 Granada, Spain; 5Généthon, Integrare Research Unit UMR_S951, INSERM, Université Paris-Saclay, Univ Evry, 91002 Evry, France; mdvmontalvoromeral@genethon.fr (V.M.-R.); gronzitti@genethon.fr (G.R.); 6Departamento de Bioquímica y Biología Molecular 3 e Inmunología, Facultad de Medicina, Universidad de Granada, Avda. de la Investigación 11, 18071 Granada, Spain; 7Departmento de Biología Celular, Facultad de Ciencias, Universidad de Granada, Campus Fuentenueva, 18071 Granada, Spain

**Keywords:** Pompe disease, lentiviral vectors, cellular disease models, optimised GAA (acid alpha-glucosidase), CRISPR/Cas9 technology, adeno-associated virus

## Abstract

Pompe disease (PD) is a rare disorder caused by mutations in the acid alpha-glucosidase (GAA) gene. Most gene therapies (GT) partially rely on the cross-correction of unmodified cells through the uptake of the GAA enzyme secreted by corrected cells. In the present study, we generated isogenic murine GAA-KO cell lines resembling severe mutations from Pompe patients. All of the generated GAA-KO cells lacked GAA activity and presented an increased autophagy and increased glycogen content by means of myotube differentiation as well as the downregulation of mannose 6-phosphate receptors (CI-MPRs), validating them as models for PD. Additionally, different chimeric murine GAA proteins (IFG, IFLG and 2G) were designed with the aim to improve their therapeutic activity. Phenotypic rescue analyses using lentiviral vectors point to IFG chimera as the best candidate in restoring GAA activity, normalising the autophagic marker p62 and surface levels of CI-MPRs. Interestingly, in vivo administration of liver-directed AAVs expressing the chimeras further confirmed the good behaviour of IFG, achieving cross-correction in heart tissue. In summary, we generated different isogenic murine muscle cell lines mimicking the severe PD phenotype, as well as validating their applicability as preclinical models in order to reduce animal experimentation.

## 1. Introduction

Pompe disease (PD), also known as acid maltase deficiency or glycogen storage disease type II (GSDII) (OMIM# 232300), is a rare autosomal metabolic disorder caused by a functional deficiency of the acid-alpha glucosidase (GAA) protein. The absence of GAA activity results in an accumulation of non-glycogen [1] in multiple tissues that ultimately leads to lysosomal dysfunction. This accumulation of glycogen also causes autophagic abnormalities [2,3,4,5]. As a consequence of these malfunctions, PD patients present a wide spectrum of phenotypes depending on the remaining levels of GAA activity. Traditionally, PD has been classified as infantile-onset Pompe disease (IOPD), with less than 2% of normal GAA activity, and late-onset Pompe disease (LOPD), with 2–40% of GAA activity. In IOPD patients, symptoms appear within their first year of life, and the disease is characterised by prominent hypotonia, muscular weakness, motor delay, feeding problems, as well as heart and respiratory failures [6,7].

The GAA protein is synthesised as a 110 kDa precursor, and most of it is transported to the lysosome, where it is processed into its most active forms (70 and 76 kDa) [8,9]. A portion of the GAA precursor is secreted and can be taken up by surrounding cells via the cation-independent mannose 6-phosphate receptor (CI-MPR). The glycosylation in the endoplasmic reticulum (ER) is crucial for GAA transport to the Golgi, where it acquires seven mannose 6-phosphate (M6P) lysosomal targeting signals required to bind to the CI-MPRs [1]. Only 10–20% of CI-MPRs are found at the cell surface [8,10], while the rest cycle between the trans-Golgi network and the endosomes. Therefore, secreted GAA precursors entering into other cells through CI-MPRs can be converted into mature GAA forms after reaching the lysosomes [11]. In fact, this cross-correction plays an important role in achieving therapeutic efficacy, not only for PD, but also for other metabolic diseases [5,12].

A total of 2075 (1205 exonic, 870 intronic) GAA variants (chromosome 17q25) have been reported in 1079 patients [13,14] (Pompe Center at Erasmus MC database), obtaining different levels of expression and/or activity of GAA proteins. The mutations are spread throughout the GAA gene and affect any of the multiple steps involved in the synthesis, post-translational modifications, lysosomal trafficking and proteolytic processing of GAA [15]. Although most GAA mutations are punctual or are found in a small number of families, there are several mutations such as c.525delT, c.1935C>A, c.2560C>T and c.2481+102_2646+31del that are frequently associated with IOPD [13,16,17].

The only current treatment available for PD is enzyme replacement therapy (ERT), which consists of the administration of exogenous rhGAA (alglucosidase alfa, Lumizyme^®^ within the USA (Sanofi Genzyme, Cambridge, MA, USA) and Myozyme^®^ outside of the USA (Sanofi Genzyme, Cambridge, MA, USA). Although ERT increases the survival of IOPD patients and stabilises the cases of the LOPD forms, it presents several limitations: (**1**) the treatment cost of this life-long therapy, (**2**) the generation of an immune response against the recombinant protein [18,19], (**3**) the lack of effect in the central nervous system (CNS) and (**4**) the progressive deterioration in skeletal and respiratory functions after years of treatment [20].

Gene therapy (GT) has emerged as an alternative treatment for PD. The two main strategies investigated for Pompe GT are the in vivo administration of adeno-associated virus (AAV) vectors and the transplantation of hematopoietic stem and progenitor cells (HSPC) transduced with lentiviral vectors (LV) expressing GAA transgene. The delivery efficiency is especially important for certain conditions such as PD. Targeting AAVs to the heart, diaphragm [21,22] or liver [23,24] has demonstrated the ability to improve correction by improving GAA delivery to most of the affected muscles [25] or by reducing the immune response and widespread distribution of GAA [24]. As an alternative to direct inoculation of AAVs, the genetic modification of HSPCs has been investigated in order to generate cellular factories that secrete GAA in affected tissues [26]. The potential advantages of this strategy include the induction of tolerance to GAA and the ability of HSPCs to differentiate into microglia at the CNS [27,28].

Cellular models for PD have traditionally been based on the isolation of primary myoblasts from human patients [29], GAA-KO mouse models [30,31] or derived from iPSCs from Pompe patients [31,32,33]. Although they are fundamental for disease mechanisms studies, these cellular model systems are difficult to culture and work with [29,34,35]. As a complementary toolbox to these cellular models, different groups have generated immortalised cell lines derived from GAA-KO mice [30,31]. However, these immortalised cellular models replicate lysosomal but not autophagic pathology [30,36]. In addition, they do not harbour analogous PD-patient mutations that could also help to understand the basic mechanism involved in the disease. Moreover, most preclinical studies use human GAA in murine cells or in mice, potentially modifying the outcome due to inter-species differences.

In the present study, we used CRISPR/Cas9 tools to generate isogenic murine GAA-KO cell lines resembling severe mutations found in Pompe patients, and validated these cells as models for PD. Then, we designed different chimeric murine GAA proteins harbouring (**1**) the IFNβ-1 leader peptide to improve secretion (IFG), (**2**) the IFNβ-1 leader peptide plus a lysosomal signal peptide (LSP) to improve secretion and lysosomal targeting (IFLG) and (**3**) the fusion protein of insulin-like growth factor 2 (2G) to allow glycosylation-independent lysosomal targeting [37]. We performed a phenotypic rescue analysis on our generated models by using lentiviral vectors, which expressed different chimeras, with the IFG chimera demonstrating the best restoration of GAA activity and normalisation of the autophagic marker p62 and surface levels of CI-MPRs. The in vivo administration of liver-directed AAVs expressing different chimeras confirmed superior behaviour of IFG and validated our new cellular models as easy-to-use tools to be included in the Pompe toolbox in order to minimise animal experimentation.

## 2. Results

### 2.1. Generation of GAA-KO Isogenic Murine Muscle Cell Models by Genome Editing

In order to generate relevant murine cellular models, we looked for the most severe mutations listed at the Pompe Center database (Erasmus MC), which catalogues mutations identified worldwide depending on their localisation and severity (www.pompevariantdatabase.nl) (accessed on 26 January 2022), and selected the most severe human mutations conserved in mouse *Gaa* gene (Figure 1A, top). Because skeletal muscle is one of the main targets for GT of PD, we used Sol8 cells (a murine skeletal muscle cell line) as the host cell line (Figure 1A, top). The selected mutations are described by the Erasmus Medical Centre database as severe because they result in truncation, frame shifts that generate aberrant proteins or inactivation of catalytic sites. In addition, all these mutations are associated with early IOPD.

We designed different gRNAs (g1, gCys, gE5 and gE7) targeting the murine loci corresponding to the selected human mutations (Figure 1A, second-top and Appendix A). Sol8 cells were nucleofected with different ribonucleoparticles (RNPs) that contained Cas9 and the different guide RNAs (Figure 1B). The *Gaa* editing efficiency ranged from 30% to 35% using the T7EI assay to measure it (Figure 1C, bottom graph, grey bars), and from 44% to 87% using the ICE algorithm (Figure 1C, bottom graph, black bars). These discrepancies (stronger in E2_bulk and E7_bulk) are due to the high frequency of insertions (E2_bulk) and deletions (E7_bulk) of one nucleotide, which are detected by the ICE algorithm but not with the T7EI assay (Figure 1D, right graphs). From each *Gaa*-edited bulk population, four to five clones were isolated and characterised (Appendix A, left). One representative clone of each mutation was selected (Appendix A, right) and named according to the Pompe mutation intended and the clonal number: ∆ ATG_14 (mutation in exon 2 targeting the ATG), E2_1 (mutation in exon 2), E5_6 (mutation in exon 5) and E7_1 (mutation in exon 7) (Figure 1E). All selected clones were confirmed to have bi-allelic targeting by means of cDNA sequencing (Figure 1E and Appendix A). Next, we characterised the effect of the different mutations on the primary structure (Figure 1F) and the predicted tertiary structure (Figure 1G) of the resulting polypeptides. The mutations in E2_1 and E5_6 clones resulted in premature stop codons, generating truncated proteins that lack the catalytic domain. The E7_1 clone has a six-nucleotide deletion that maintains the full-length GAA polypeptide, but lacks two amino acids (393Arg and 394Thr) which are critical for the GAA activity [38] (Appendix A). The karyotypes of the different clones were not altered (Appendix A) and no significant off-target was found (Appendix A).

### 2.2. Murine GAA-KO Sol8 Clones Lack GAA Activity, Present Increased Autophagy Markers and Glycogen Content as Well as Surface Downregulation of Mannose 6-Phosphate Receptors (CI-MPRs)

All isogenic Sol8 cellular models generated for each mutation (∆ ATG_14, E2_1, E5_6 and E7_1) showed a partial reduction in GAA mRNA (Figure 2A) and almost a complete lack of GAA activity (Figure 2B).

The loss of GAA activity in Pompe patients results in glycogen accumulation and impaired autophagy, causing muscle deterioration [39]. We therefore analysed whether our muscle cellular models could mimic these phenotypic hallmarks of PD. The accumulation of glycogen in gene edited bulk populations was variable (Appendix A), probably due to differences in genome editing efficacy (see Figure 1C, bottom, black bars). All selected clones have a tendency to accumulate glycogen upon myotube differentiation compared with WT, but only ∆ATG_14 and E7_1 glycogen deposits were significant (Figure 2C and Appendix A). Similarly, p62, a hallmark of impaired autophagy [40] and a marker for disease progression in PD [41], was also increased in clones and bulk populations before (Figure 2D, left-bottom panel and Figure 2E, white bars) and after (Figure 2D, right-bottom panel and Figure 2E, black bars) differentiation into myotubes. In this direction, the lysosomal marker LAMP-1 presented a trend to be increased in the generated clones after differentiation (Figure 2D, top panels and Figure 2F, black bars). The elevated p62 and LAMP-1 are indicative that autophagic buildup is occurring in our cellular models, although not all clones reached statistical significance.

Cardone et al. previously showed reduced levels of CI-MPRs at the surface of fibroblasts from Pompe patients [42]. We therefore analysed whether the different GAA-KO cellular models mimic this phenotypic defect. Although the mRNA expression levels were equivalent in wild-type (WT) Sol8 and in the selected clones, E2_1, E5_6 and E7_1 (Figure 2G), a clear decreased trend in surface CI-MPRs levels was observed in all clones and bulk populations, being especially relevant in the E2_1 clone (Figure 2H,I).

### 2.3. Design of Different Chimeric GAA Proteins for Gene Therapy Approaches

Next, we designed three different GAA chimeras based on the codon-optimised sequence of murine cDNA (Figure 3A): 1—the IFG protein, containing the leader peptide of the murine IFNβ1 at the N-terminus, which should increase mGAA secretion; 2—the IFLG protein that contains, in addition to the IFNβ1 leader peptide, four lysosomal signal peptides (LSPs, 2 YQRLC and 2 CNPGY) separated by GAG linkers (Figure 3A) and 3—the 2G protein that incorporates the leader peptide and a 7-68 polypeptide of mature mIGF2 (Figure 3A), designed to keep binding for M6PR/IGF2R [43] while reducing binding to IGFIR and IGFBP [43,44,45].

The hydropathicity pattern prediction using the Kyte–Doolittle scale in ExPASy [46,47] showed a very similar hydrophobicity pattern of the different mGAA chimeras compared to the original mGAA (Figure 3B). Additionally, a theoretical reconstruction of the different mGAA proteins using I-TASSER prediction, Pymol and Chimera software showed that all chimeric proteins preserved the domains involved in GAA activity without significant differences in the 3D structure of the GAA protein (Figure 3C). A scheme of the potential therapeutic advantages in the secretion and uptake of the different chimeric mGAA proteins for GT strategies can be seen in Figure 3D.

### 2.4. Murine Cellular Models Point to IFG as the Best Chimera in Terms of Expression, Secretion and Restoration of PD Defects

To study the performance of the different GAA chimeras for potential GT application, we generated different lentiviral vectors (LVs) expressing IFLG, IFG and 2G under the SFFV promoter (Spleen Focus Forming Virus) (SIFLG, SIFG and S2G, respectively; Figure 4A).

The E2_1 GAA-KO cellular model was transduced with these different LVs (the vector copy number per cell in each population is indicated in Appendix A) to investigate the behaviour of the different GAA chimeras restoring the GAA activity (Figure 4B, graph and middle panel) and secretion level of the optimised GAA (Figure 4B, right panel). E2_1 clone transduced with LVs expressing the IFG chimera (SIFG) showed the highest GAA restoration and secretion activities (Figure 4B, graph), despite the fact that SIFLG showed higher levels of processed protein (Figure 4B, middle panel). This could be due to the lower activity of the SIFLG protein as a consequence of the insertion of the LSP. As expected, SIFG and SIFLG were mostly processed to the active forms of 70–75 kDa intracellularly (Figure 4B, middle panel) and secreted as unprocessed 110 kDa protein (Figure 4B, right panel). However, in E2_1 cells, most of the 2G chimeras were unprocessed and were retained intracellularly with poor secretion (Figure 4B and Appendix A). Since the 2G remains largely unprocessed, its ability to restore GAA activity in E2_1 cells was also reduced in comparison to the other chimeras (Figure 4B, left graph). We next analysed the therapeutic potency of the different GAA chimeras in the E2_1 cellular model. Interestingly, all LVs could normalise p62 levels (Figure 4C and Appendix A), although only LVs expressing SIFG showed a partial improvement restoring normal surface levels of CI-MPRs (Figure 4D).

To identify whether potential differences in processing/secretion were cell-dependent, we investigated the expression, processing and secretion of the different GAA chimeras in myeloid (RAW 264.7) and hepatic (Hepa 1–6) murine cell lines. Interestingly, although the expression and processing of the different chimeras were similar in all cell lines (Figure 4B,E,F, left graphs and Appendix A), only Hepa 1-6 efficiently secreted all chimeras, including the 2G protein (Figure 4F, right panel).

### 2.5. Human Cellular Models Uncover Crucial Differences in Secretion of the 2G Chimera Compared to Murine Cells

We next investigated the behaviour of the murine GAA chimeras in K562 (myelogenous leukaemia—lymphoblast) Meg-01 (myelogenous leukaemia—megakaryoblast) and SJCRH30 (rhabdomyosarcoma—muscle cells) human cell lines in order to determine potential cross-functions of the different mouse-derived domains (IFLG, IFG and 2G) in human cells. To this end, these cell lines were transduced with the different therapeutic LVs, and GAA protein expression and activity were analysed as previously described (Figure 5A–C).

We were not able observe significant differences in intracellular alfa-glucosidase activity nor in GAA processing (left graphs and Western blot (middle panels) of Figure 5A–C) compared with murine cell lines (left graphs and Western blot (middle panels) of Figure 4B,E,F), since, in murine cells, both myeloid (Figure 5A,B) and muscle (Figure 5C) human cells transduced with the SIFG LVs achieved their highest intracellular activity in relation to their vector copy number (Appendix A). Surprisingly, the secretion of the 2G protein (cells transduced with S2G LVs) was easily detected in all human cell lines (Figure 5A–C; Western blots, middle panels and Appendix A). In fact, this chimera presented the highest secretion level of all chimeras in human myeloid cells (K562 and Meg-01) and similar levels to the IFG protein in the human muscle cell line SJCRH30 (Appendix A). Therefore, these data suggest that the murine cellular models do not necessarily mimic the secretion profile of newly designed proteins in human cells.

### 2.6. Poor Cross-Correction of GAA-KO Cellular Model with GAA Chimeras

Since IFLG and 2G chimeras were designed to enhance GAA cellular/lysosomal uptake as described in Figure 3D, we analysed the efficacy of cross-correction of the IFG, IFLG and 2G proteins by measuring their functional uptake in E2_1 cells (Figure 6A).

First, we quantified the GAA activity found in the conditioned media obtained from non-transduced (NT) and transduced K562 cells with the different LVs (Appendix A). The different conditioned media were added to the E2_1 clone, and GAA activity was measured 20 h later. A significant restoration of the enzymatic activity was observed in all E2_1 cells incubated with conditioned media from transduced K562 cells (SIFLG, SIFG and S2G), but not when using media from non-transduced K562 cells (Figure 6B). However, no significant differences were observed between the different chimeras. Moreover, the GAA activity achieved in cross-corrected E2_1 cells was 5–10 times lower than in WT Sol8 cells (Figure 2B vs. Figure 6B), emphasising the difficulties in cross-correcting Pompe muscle cells. However, it is interesting to highlight that this low uptake of GAA by E2_1 cells mimics what it is observed in muscle from PD patients and is probably due, at least in part, to the low CI-MPRs levels present in GAA-KO cells.

### 2.7. In Vivo Comparison of the Different mGAA Chimeras in GAA-KO Mice

We finally analysed the performance of the different GAA chimeras for in vivo gene therapy application, focusing on their ability for cross-correction. For this aim, we generated different adeno-associated viral vectors (AAV) expressing codon-optimised murine GAA, IFLG, IFG and 2G under the expression of the human alpha 1-antitrypsin (hAAT), a human hepato-specific promoter (AG, AIFLG, AIFG and A2G, respectively; Figure 7A) and tested them in GAA-KO mice (Figure 7B).

Mice were sacrificed one month after the injection, and the vector genome copy number (Figure 7C), as well as the GAA activity in the liver (Figure 7D) and serum (Figure 7E, left graph), were analysed to determine in vivo GAA expression and secretion activity for each of the constructs. As already reported [48], the unmodified-codon optimised GAA appears as the best performer in terms of expression in the liver (Figure 7D, right graph). However, modified GAAs were secreted with similar efficacy, as evidenced by normalising the GAA activity in serum versus GAA activity in the liver (Figure 7E, right). We next analysed GAA activity and glycogen content in the heart as an indicator of the ability of the different GAAs to cross-correct muscle tissue. As it can be observed in Figure 7F, left and Figure 7G, all chimeras appear to behave similarly, restoring physiological levels of GAA activity and reducing glycogen levels, although the IFG and 2G modified GAA seem to improve the uptake compared to wild-type GAA (AG) and AIFLG (Figure 7F, right).

## 3. Discussion

The development of accurate cellular models helps preclinical studies based on the screening of new drugs or GT tools. Therefore, one of the aims of this manuscript was to generate a murine–murine cellular model to investigate rationally designed murine GAA chimeras to refine the strategy before translation into animal models. There are several in vitro murine cellular models available [31,36,49]; unfortunately, they do not fully mimic severe IOPD signatures or LOPD disease progression. Additionally, engineered GAA-KO mice generally lack human analogue mutations that mimic those described in PD patients.

Being aware of the genetic and physiological interspecific differences between the gold-standard GAA-KO model and most PD patients, we first generated murine muscle cell models using genome editing to reproduce relevant PD mutations listed in the PD database [50,51]. Interestingly, in addition to succeeding in abrogating GAA activity, all cellular models generated for this work presented several phenotypic defects characteristic of PD, such as increased autophagic build-up and glycogen accumulation as well as a downregulation of surface CI-MPRs, a phenomenon that has been reported previously only in fibroblasts from Pompe patients [42,52].

A massive accumulation of autophagic debris, also known as autophagic build-up, is a landmark of PD and contributes to the development of muscle weakness and disease progression [41]. This phenotype results in the accumulation of p62/LC3 (autophagic markers) and LAMP-1 (lysosomal marker), both of which are clearly observed in patients and animal models [5]. However, although previously described murine and human cellular models resemble glycogen accumulation [53], only a few human cellular models have been reported to mimic autophagic build-up [54]. Surprisingly enough, our murine isogenic cellular models not only accumulated glycogen, but also presented clear evidence of autophagic build-up after myotube differentiation, as indicated by increased p62 and LAMP-1.

ERT therapy and gene therapy cross-correction strategies are based on the binding of GAA to CI-MPRs, they enter into the cells through clathrin-coated vesicles and, finally, their endosomal fusion occurs in order to (**1**) deliver the GAA enzyme into the lysosomes to mature and become active and (**2**) to recycle the M6PR to the cell surface [2,55]. It has been described that the autophagic build-up disrupts normal protein trafficking. In particular, Fukuda et al. detected unprocessed GAA as well as CI-MPR proteins trapped in autophagic areas in cells from GAA-KO mice [56]. This processing block could lead, not only to lower GAA activity into the cells, but also to lower M6PR recycling as well as its surface availability. In line with this, Cardone et al. found a marked surface CI-MPR reduction that correlated with PD severity [42], pointing to defects in receptor recycling as the driver of this phenomenon. This reduction in surface CI-MPRs in Pompe patients’ cells was confirmed by several authors [52,56]. All of our murine cellular models presented lower surface expression of CI-MPRs, although this reduction was highly variable depending on culture conditions (data not shown). Therefore, our isogenic cellular models represent the first ones that allow us to investigate the effects of patients’ GAA mutations on autophagic build-up and surface CI-MPR downregulation. The reduction in membrane-bound CI-MPRs in cells lacking GAA can explain the poor responses to ERT in the most severe PD patients [1,57,58] and favours the search for new strategies focused not only on GAA restoration, but also on increasing CI-MPR surface expression, such as the use of salmeterol [59,60] or clenbuterol, two selective b2-agonists [61]. In this sense, there is an ongoing phase II clinical trial with clenbuterol in adult patients with PD stably treated with ERT (NCT04094948).

To validate the potential applicability of our new isogenic cellular models as tools for preclinical studies, we analysed the ability of three different GAA chimeras to restore their GAA activity and their phenotypic defects directly (LV- transduced cells) or by cross-correction (adding supernatant from GAA-expressing cells). Interestingly, only the SIFG LVs could partially overcome CI-MPRs surface reduction and also normalised p62 levels, indicating that IFG could be a suitable candidate for GT applications. This observation was confirmed by our in vivo analysis using liver-directed AAVs on GAA-KO mice, validating our in vitro cellular models as a tool to refine animal experimentation.

Our data of the murine IGF2-GAA (2G) chimera contrast with the results obtained by several groups using human IGF2-GAA proteins, showing clear improvements in their therapeutic activity in comparison with WT GAA [48]. The differences in IGF2-GAA processing in human versus murine myeloid cells could be a consequence of several factors. Interestingly, the mouse *Igf2* expression declines after birth [62], while human *Igf2* remains active throughout life [63]. The fact that the murine and human IGF2 share an amino acid identity of 94% (Appendix A) could explain the good secretion profile of the murine 2G chimera by human cells.

In summary, we generated new isogenic cellular models of severe PD that mimic hallmark features described in Pompe patients: a lack of GAA activity, the accumulation of glycogen, autophagic build-up and the downregulation of surface CI-MPRs.

Our isogenic murine GAA-KO models, together with the human cell lines, are useful tools for the analysis of LVs expressing different GAA chimeras. In this sense, although LV-IFG (SIFG) is a potential candidate for GT of PD, we consider that it requires further refinement before moving forward to GT applications.

## 4. Materials and Methods

### 4.1. Cell Lines and Culture Media

HEK293T cells (CRL11268; American Type Culture Collection; Rockville, MD, USA) were maintained in DMEM medium (Dulbecco’s modified Eagle medium high glucose) (Biowest, Nuaillé, France) supplemented with 10% foetal bovine serum (FBS) (Biowest) and 1% penicillin/streptomycin (Biowest) at 10% CO_2_ and 37 °C. The human cell line, K562 (lymphoblast from bone marrow chronic myelogenous leukaemia, ATCC^®^ CCL-243™), was maintained in RPMI medium (Biowest), supplemented with 10% FBS and 1% penicillin/streptomycin at 5% CO_2_ and 37 °C. The murine skeletal muscle cell line, Sol8 (ATCC^®^ CRL-2174™), was kindly provided by Dr. Francisco Hernández Torres (Fundación Medina, Granada, Spain) and it was cultured in DMEM medium (Dulbecco’s modified Eagle medium high glucose) (Biowest) supplemented with 20% FBS and 1% penicillin/streptomycin at 37 °C at 5% CO_2_. The human megakaryoblast cell line, Meg-01 (ATCC^®^ CRL-2021™), was cultured in RPMI medium supplemented with 10% FBS and 1% penicillin/streptomycin at 5% CO_2_ and 37 °C. Murine macrophages, RAW 264.7 (ATCC^®^ TIB-71™), were grown in DMEM with 10% FBS and 1% penicillin/streptomycin at 5% CO_2_ and 37 °C. Human muscle cells, SJCRH30 (ATCC^®^ CRL-2061™), were cultured in RPMI medium with 10% FBS and 1% penicillin/streptomycin at 5% CO_2_ and 37 °C. Murine hepatic cell line, Hepa 1-6 (ATCC^®^ CRL-1830™), was kindly provided by Dr. Giuseppe Ronzitti (Genethon, Evry, France) and it was cultured in DMEM medium supplemented with 10% FBS and 1% penicillin/streptomycin at 5% CO_2_ and 37 °C.

### 4.2. Generation of GAA-KO Sol8 Cell Lines by Cas9 RNP Nucleofection

We used the CRISPOR design tool (CRISPOR.org, accessed on 30 April 2022) [64] for our different gRNAs, the sequences of which are as follows: g1: GAGGGGCTTCCGTATATTCA; gCys: CATCTCACAGGAGCAATGCG; gE5: TTGCTAAACAGCAATGCCAT; gE7: AGGTAGTGGAGAACATGACC. For the formation of the RNP, chemically synthesised crRNA and tracrRNA obtained from Synthego (Silicon Valley, CA, USA) (200 μM) were mixed and incubated following the manufacturer’s instructions to form guide RNA (gRNA) at a concentration of 30 μM. Next, this gRNA was mixed in a 1:3 ratio by volume with high-fidelity Cas9 (IDT, Coralville, IA, USA) and incubated at room temperature for 15 min to form RNP. For gene disruption, Sol8 cells (2 × 10^5^) (passage 8) were nucleofected with RNP. Nucleofection was performed with an Amaxa Nucleofector 4-D and solution SF cell line (Lonza, Basel, Switzerland), applying programme CD-137 in 20 μL Nucleocuvette™ Strips.

### 4.3. Quantification of Cleavage Efficiency of GAA Target Site

Genomic DNA extraction from Sol8 cells was performed with a QIAamp DNA mini kit (Qiagen, Hilden, Germany). The genomic regions flanking the CRISPR target site for each treatment were amplified by PCR using different pairs of primers with KAPA Taq PCR (Kapa Biosystems, Wilmington, MA, USA): GAA exon 2 Fw: AAGATGCTCTGGCTGCCT and GAA exon 2 Rev: TGCTCTGCCTAGCCTGTC for ∆ ATG and E2 cells; GAA Exon 4 Fw: AGTTCCTGCAGCTGTCCA and GAA Intron 6 Rev: AAGTGTTTGGGCTCAGGAA for E5 cells; GAA Exon 6 common Fw: TCCTGAGCCCAAACACTTCT and GAA Exon 8 Rev: CCACGATCATCATGTAGCG for E7 cells. The fragments were purified with a QIAquick PCR product purification (Qiagen), in accordance with the manufacturer’s protocols. When PCR products were purified, 200 ng was mixed with 2 μL of T7 buffer (10X NEBuffer (New England Biolabs, Ipswich, MA, USA) and ultrapure water to a final volume of 20 μL, and subjected to a reannealing process to enable heteroduplex formation. After reannealing, products were treated with T7 endonuclease I (New England Biolabs) for 1 h at 37 °C. Then, we analysed the samples on 2% agarose gels. The percentage of cleavage was determined by densitometry of the bands, using the following formula: $\mathrm{Indel}\;\left(\%\right)=100\times [1-{\left(1-\mathrm{fraction}\;\mathrm{cleaved}\right)}^{1/2}]$, where the fraction cleaved is = $\frac{\mathrm{Density}\;\mathrm{of}\;\mathrm{cleaved}\;\mathrm{DNA}}{\mathrm{Density}\;\mathrm{of}\;\mathrm{uncleaved}\;\mathrm{DNA}\;+\;\mathrm{Density}\;\mathrm{of}\;\mathrm{cleaved}\;\mathrm{DNA}}$

### 4.4. ICE

Genomic DNA from Sol8 cells was extracted 5 days after nucleofection using a QIAamp DNA mini kit (Qiagen) following the manufacturer’s instructions. The genomic regions flanking the CRISPR target site for each treatment were amplified by PCR using different pairs of primers with KAPA Taq PCR (Kapa Biosystems): GAA exon 2 Fw: AAGATGCTCTGGCTGCCT and GAA exon 2 Rev: TGCTCTGCCTAGCCTGTC for ∆ ATG and E2 cells; GAA Exon 4 Fw: AGTTCCTGCAGCTGTCCA and GAA Intron 6 Rev: AAGTGTTTGGGCTCAGGAA for E5 cells; GAA Exon 6 common Fw: TCCTGAGCCCAAACACTTCT and GAA Exon 8 Rev: CCACGATCATCATGTAGCG for E7 cells. The fragments were purified with the QIAquick PCR product purification kit (Qiagen), in accordance with the manufacturer’s protocols. For analysing allele modification frequencies, we used ICE (Inference of CRISPR Edits), a web-based analysis tool developed by Synthego (https://ice.synthego.com/#/, accessed on 20 October 2019) [65]. Our purified PCR products were Sanger-sequenced using both PCR primers. Then, each sequence chromatogram was analysed with the ICE software (Synthego, Silicon Valley, CA, USA). Analyses were performed using a control sequence. The ICE score showed editing by NHEJ.

### 4.5. Off-Target Analysis

The top listed off-targets were selected from the CRISPR designed tool (Synthego) for the different Sol8 GAA-KO clones. Off-target analysis were performed by ICE analysis, as previously described. The genomic regions flanking the CRISPR target site for each off target at the different clones were amplified by PCR using different pairs of primers (Appendix A).

### 4.6. Plasmid Design

First, the murine GAA sequence was codon optimised by GenScript (Piscataway, NJ, USA). The 2G sequence was as follows:

MGIPVGKSMLVLLISLAFALCCIAALCGGELVDTLQFVCSDRGFYFSRPSSRANRRSRGIVEECCFRSCDLALLETYCATPAKSEGAP, containing the signal peptide, the IGF2 sequence without six amino acids which presented protease cutting sites and two spacer amino acids. Optimised mGAA was added after this sequence. The IFG construct contained the interferon beta 1 leader peptide: MNNRWILHAAFLLCFSTTALS followed by optimised mGAA. IFLG contained the interferon beta 1 leader peptide followed by lysosomal sorting peptides (LSP) described by Dekiwadia et al. [66]. Between LSPs, spacers and flexible amino acids were added:

MNNRWILHAAFLLCFSTTALSYQRLCGGACNPGYGAGYQRLCGGACNPGYAI. Optimised mGAA was added after this sequence. All sequences were obtained from the UniProtKB database [67]. Predictions of hydrophobicity patterns were performed through the online database ExPASy Bioinformatics Resources Portal [68] and the Kyte–Doolittle hydropathy scale [46].

### 4.7. Lentiviral Constructions

The lentiviral plasmid SIFLG was obtained by the incorporation of the sequence IFLG in the place of eGFP in a 2nd-generation lentiviral backbone, SEWP. pUC57 plasmid (GenScript) containing the IFLG sequence and SEWP were digested with BamHI and Sbf1 (New England Biolabs) and the resulting plasmids were ligated with T4 DNA ligase (New England Biolabs). The lentiviral plasmid SIFG was obtained by cloning the sequence IFG in the place of IFLG in the SIFLG lentiviral plasmid. pUC57 plasmid (GenScript) containing the IFG sequence and SIFLG was digested with BamHI and AsiSI (New England Biolabs), and the resulting plasmids were ligated with T4 DNA ligase (New England Biolabs). The lentiviral plasmid S2G was obtained by cloning the sequence 2G in the place of IFLG in the SIFLG lentiviral plasmid. pUC57 plasmid (GenScript) containing the 2G sequence and SIFLG was digested with PacI and AsiSI (New England Biolabs), and the resulting plasmids were ligated with T4 DNA ligase (New England Biolabs). The lentiviral plasmid SG was obtained after the digestion of S2G plasmid with AsiSI (New England Biolabs) in order to eliminate the IGF2 cassette and the resulting plasmid was ligated with T4 DNA ligase (New England Biolabs). After the different ligations and transformation into E. coli Stbl3 competent bacteria (Life Technologies, Thermo Fisher Scientific, Waltham, MA, USA), the plasmids were obtained using Wizard^®^ Plus SV Minipreps DNA (Promega, Madison, WI, USA). The restriction pattern was performed and the whole plasmid was eventually sequenced. Maxi-production was performed using NucleoBond^®^ Xtra Maxi (Macherey-Nagel, Düren, Germany).

### 4.8. Vector Production and Virus Titration

LV particles were produced by polyethylenimine (PEI) (408727, Sigma-Aldrich, St. Louis, MO, USA), as previously described [69]. Briefly, for second-generation LVs, 293T packaging cells were transfected with packaging plasmid pCMvdR8.91, plasmid pMD2.G.47 encoding the vesicular stomatitis virus (VsV-g) envelope gene (http://www.addgene.org/Didier_Trono/, accessed on 2 June 2020) and the desired vector plasmid (SG, SIFLG, SIFG or S2G). The producer cells were cultured for 24, 48 and 72 h and the viral supernatants were collected and filtered through 0.45 μm filters (Nalgene, Rochester, NY, USA). The viral particles were then concentrated by ultracentrifugation in a Beckman Optima Centrifuge (Beckman Coulter, CA, USA) at 40,000 rpm for 2 h at 4 °C, and the viral pellets were resuspended in StemSpan medium (StemCell Technologies, Vancouver, Canada) for 1 h on ice, aliquoted, and immediately frozen at −80 °C.

Viral titres (transduction units [TU]/mL) were calculated using quantitative PCR. Briefly, 10^5^ K562 cells were transduced with serially diluted amounts of LV. Genomic DNA was isolated (10^5^ cells, equivalent to 0.6 μg of genomic DNA) (kit QiAamp DNA Mini Kit) (Qiagen) and the copy number of LVs integrated was measured using a standard curve (from 10^5^ to 10 copies) of plasmid DNA. We used KAPA SYBR FAST Universal qPCR (KAPA Biosystems) in a Mx3005P QPCR System from Stratagene (Agilent Technologies, Santa Clara, CA, USA). The primers used for titration were ∆U3 (fw: GACGGTACAGGCCAGACAA) and PBS (rev: TGGTGCAAATGAGTTTTCCA).

### 4.9. Cell Transduction

Around 10^5^ cells were plated in 48-well plates (BD Biosciences, San Jose, CA, USA) and incubated with different MOIs (5, 20 and 50) of virus. After 5 h, cells were washed and maintained in culture for further analysis.

### 4.10. AAVs Constructions and Vectors (Design and Production)

Transgene sequences were cloned into an AAV vector backbone under the transcriptional control of the apolipoprotein E (hepatocyte control region enhancer) and the human alpha 1-antitrypsin (hAAT) promoter, as described in [70,71]. We designed and cloned an oligo with PacI and SbfI restriction sites in order to clone our mGAA chimeras from our lentiviral vectors (SIFLG, SIFG, S2G and SG) in an AAV vector backbone. Then, we digested with PacI (New England Biolabs) and SbfI (New England Biolabs) our lentiviral vectors (SIFLG, SIFG, S2G and SG) and the AAV vector backbone. The resulting plasmids were ligated with T4 DNA ligase (New England Biolabs). After the different ligations and transformations into *E. coli* Stbl3 competent bacteria (Life Technologies, Thermo Fisher Scientific, Waltham, MA, USA), the plasmids were obtained using Wizard^®^ Plus SV Minipreps DNA (Promega, Madison, WI, USA). The restriction pattern was performed and the whole plasmid was eventually sequenced. Maxi-production was performed using NucleoBond^®^ Xtra Maxi (Macherey-Nagel, Düren, Germany).

The research-grade AAV vectors used in this study were produced using an adenovirus-free transient transfection method. Briefly, suspension HEK293 cells were transfected using PTG1-plus (POLYTHERAGENE) with the three plasmids containing the adenovirus helper proteins, the AAV Rep and Cap genes, and the ITR-flanked transgene expression cassette. Then, 24 h after transfection, cells were treated with Benzonase^®^ (Merck-Millipore, Darmstadt, Germany), and 2 days later, they were lysed with Triton (Sigma, St Louis, MO, USA) and clarified by filtration. Vectors were then purified by a single immunoaffinity chromatography column, using POROS CaptureSelect (Thermo Fisher Scientific, Waltham, MA, USA) resins. Purified particles were formulated in phosphate buffered saline containing 0.001% of Pluronic F68 (Sigma Aldrich, Saint Louis, MO, USA), and stored at −80 °C. Titres of AAV vector stocks were determined using quantitative real-time PCR (qPCR). Specific primers were as follows: forward 5′-GGCGGGCGACTCAGATC-3′, reverse 5′-GGGAGGCTGCTGGTGAATATT-3′.

### 4.11. In Vivo Studies

Mice studies were performed according to French and European legislation on animal care and experimentation (2010/63/EU) and approved by the local institutional ethical committee (protocol no. 2015-008). Gaa^−/−^ mice were purchased from the Jackson Laboratory (B6;129-Gaatm1Rabn/J, stock no. 004154, 6neo) and were originally generated by Raben and colleagues [72]. AAV vectors were administered intravenously via the tail vein in a dose of 5 × 10^11^ vg/kg, five mice per group. One month after the treatment, all the mice were sacrificed for tissue analysis.

### 4.12. Vector Copy Number (qPCR)

We used genomic DNA of 1 copy per cell to perform a standard curve to calculate the vector copy number. We used KAPA SYBR FAST Universal qPCR (KAPA Biosystems) in a programme (95 °C × 3′’ + 60 °C × 30′’) × 40 cycles + (95 °C × 1′ + 55 °C × 30′’ + 95 °C × 30′) Mx3005P QPCR System from Stratagene (Agilent Technologies, Santa Clara, CA, USA). The sequences of the primers were ∆U3 (fw: GACGGTACAGGCCAGACAA) and PBS (rev: TGGTGCAAATGAGTTTTCCA).

### 4.13. GAA Secretion

We plated 10^7^ control (NT, non-transduced) and transduced (SIFLG, SIFG y S2G) suspension cells (K562 and Meg-01) in a 48-well plate and 3 × 10^6^ control (NT) and transduced (SIFLG, SIFG and S2G) adherent cells (Sol8, RAW 264.7, Hepa 1–6 and SJCRH30) in a 6-well plate in 1 mL of Opti-MEM with 1% P/S (Thermo Fisher Scientific) for 16 h at 37 °C and 5% CO_2_. Supernatants were centrifuged at 13,000 rpm, 4 °C for 15 min. GAA activity (see GAA activity protocol) and expression (Western blot) were analysed in the supernatant. Briefly, for protein precipitation, we added 1/4 volume of chloroform and 1 volume of methanol, and vortexed the mixture. After centrifugation at 12,000 rpm for 5 min at RT, we eliminated the interphase of methanol and added 1 volume of methanol for a second centrifugation. Finally, the pellet was allowed to dry at 50 °C for 5–10 min and analysed by means of Western blot.

### 4.14. Western Blot

A total of 3 × 10^6^ cells, previously washed in PBS, precipitated protein from supernatant cell culture, or animal tissues were lysed in 90 μL, 30 μL or 250 μL of RIPA buffers, respectively: RIPA Lysis Buffer with 100X protease inhibitors (Thermo Fisher Scientific). The lysates were incubated on ice for 1 h. The samples were centrifuged at 13,000 rpm for 15 min at 4 °C and the supernatant was collected in 4x Laemmli Sample Buffer (Bio-Rad, Hercules, CA, USA). Samples were loaded on a 7–14% gradient acrylamide gel (Bio-Rad), using 10× Tris/Glycine/SDS as a running buffer and the Bio-Rad electrophoresis system. The gels were transferred to PVDF membranes and blocked with 5% milk in TBS-Tween for 1 h. The membranes were incubated with different primary antibodies: anti-GAA (A7674-50, Abclonal Woburn, MA, USA), anti-p62 (18420-1-AP, Proteintech, Rosemont, IL, USA), anti-LAMP-1 (14-1071-82, Invitrogen Carlsbad, CA, USA) and anti-ERK1,2 (06-182A, Merck Millipore, Burlington, MA, USA) at 4 °C overnight and incubated with the secondary antibodies: goat anti-rabbit IRDye^®^ 800CV and donkey anti-rabbit IRDye^®^ 700CV (LI-COR) for 1 h. The membranes were scanned with the Odissey^®^ system (LI-COR, Lincoln, NE, USA).

### 4.15. GAA Activity Assay (Intracellular and Extracellular)

We measure GAA activity using a fluorometric kit (Lysosomal alpha-Glucosidase Activity Assay kit) (Abcam, Cambridge, UK) based on 4-methylumbelliferyl-α-D-glucopyranoside (4-MUG). GAA hydrolyses 4-methylumbelliferyl α-D-glucopyranoside and releases 4-methylumbelliferone (4-MU) that can be measured fluorometrically [73,74]. Briefly, 10^6^ cells or 10 mg of animal tissue were lysed in 150 μL of the GAA assay buffer for 20 min on ice. After that, cells were centrifuged at 12,000 rpm for 5 min at 4 °C and 10 μL of the supernatant was placed in a 96-well plate. Protein quantification was performed using Pierce BCA Protein Assay Kit (Thermo Fisher Scientific). At the same time, a standard curve was prepared using the standard of 4-methylumbelliferone at 100 μM (Abcam) following the protocol indications and adding GAA assay buffer until gaining a volume of 60 μL per well. We added 40 μL of GAA assay buffer in a separate well as a background control. Finally, 20 μL of substrate (Abcam) was added into all wells, except standard curve wells, and incubated at 37 °C for 90 min, protected from light. The reaction was stopped by adding 100 μL of stop buffer (Abcam) and fluorescence intensity (λex = 368 nm and λem = 460 nm) was measured by using a fluorescence microtitre plate reader (Infinite 200 PRO NanoQuant) (Tecan, Männedorf, Switzerland).

### 4.16. Myotubes Differentiation

For PAS staining, 18,000 Sol8 cells were cultured in 4-well slide chambers (Nunc, Roskilde, Denmark) in DMEM high glucose (supplemented with 20% FBS and 1% P/S). For glycogen content measurement, 250,000 Sol8 cells were cultured in 6-well plates in DMEM high glucose (supplemented with 20% FBS and 1% P/S). After 48 h, the medium was replaced with DMEM high glucose (supplemented with 1% of FBS and 1% P/S). At day 6, a 9-h starving was performed with DMEM without glucose (Gibco, Amarillo, TX, USA). Cells were collected by lysing in PBS-triton with protease inhibitor (1X) for glycogen detection.

### 4.17. PAS Staining

The staining was performed using Atrys (Granada, Spain). Briefly, slides were washed with PBS, fixed with 4% of p-formaldehyde and incubated in a 0.5% periodic acid solution for 25 min. After washing, they were stained with Schiff´s reagent for 30–40 min. The contrast was performed with haematoxylin for 10 s (all reagents from Merck/Sigma-Aldrich). Slides were dehydrated and mount using DPX. Images were obtained with an Olympus upright BX43 microscope (10× objective).

### 4.18. Glycogen Quantification

Glycogen content was measured as described in by Cagin, U. et al. [75]. The glycogen was indirectly measured whilst the glucose was released after total digestion with amyloglucosidase from *Aspergillus niger* (Merck Life Science, Darmstadt, Germany). Sol8 myoblasts were differentiated into myotubes, as previously described. Lysates from cells and animal tissues were incubated for 5 min at 95 °C and then cooled at 4 °C; 25 μL of amyloglucosidase diluted at 1:5 in 0.1M potassium acetate (pH 5.5) was added to each sample. A control reaction without amyloglucosidase was prepared for each sample. Both sample and control reactions were incubated at 37 °C for 90 min. The reaction was stopped by incubating samples for 5 min at 95 °C. The released glucose was determined using a glucose assay kit (Merck Life Science) by measuring absorbance with Infinite 200 PRO NanoQuant (Tecan, Männedorf, Switzerland) at 540 nm.

### 4.19. RNA Extraction

Total RNA was obtained using the Trizol reagent (Thermo Fisher Scientific). Briefly, samples were incubated for 5 min at RT, 200 μL of chloroform was added, mixed, and incubated for 3 min at RT. Samples were centrifuged in Phase Lock Gel Heavy tubes (VWR, Radnor, PA, USA) at 12,000× *g* at 4 °C for 15 min. Afterwards, the aqueous phase was collected to precipitate RNA by adding 450 μL of isopropanol 100%. After centrifugation at 12,000× *g* at 4 °C for 10 min, the pellet was washed with ethanol 75%, resuspended in RNase free water and kept at −80 °C until further use.

### 4.20. RT-PCR

RNA samples were reverse transcribed using the Superscript first-strand system (Thermo Fisher Scientific). Afterwards, we conducted a qPCR with primers for GAA cDNA (fw: TACGCAGGAGGTCGTGT and rev: GTCTGCTCCTGGATGTGC) to amplify the amplicon of 372 bp with KAPA Taq PCR (Kapa Biosystems). The fragment was purified with a QIAquick PCR product purification (Qiagen), in accordance with the manufacturer’s protocols, and we sequenced it using Sanger. The results were analysed using the Basic Local Alignment Search Tool (BLAST) and ICE software (Synthego). CI-MPRs RNA analyses were conducted using qPCR with TaqMan™ Gene Expression Assay (FAM) (4453320, inventoried Assay ID: Mm00439576_m1) (Thermo Fisher Scientific) following the manufacturer’s instructions.

### 4.21. GAA Uptake Assay

A total of 2 × 10^7^ control and transduced K562 cells (NT, SIFLG, SIFG and S2G) were plated in 24-well plates in 2 wells with 1.5 mL/well of Opti-MEM with 1% P/S (Thermo Fisher Scientific). In parallel, 1.75 × 10^5^ E2_1 cells were plated in a 6-well plate. After 20 h, the conditioned medium from K562 cells was centrifuged at 13,000 rpm for 15 min at 4 °C and supernatant was added to E2_1 cells. After 20 h, GAA intracellular activity was measured.

### 4.22. Surface CI-MPRs Analysis by Flow Cytometry

A total of 2.5 × 10^5^ Sol8 cells (WT, bulk populations, E2_1 and transduced E2_1 cells) were plated in 6-well plates with 2 mL of DMEM (supplemented with 20% FBS and 1% penicillin/streptomycin). After 24 h, the cells were detached using cell scrapers and kept on ice (4 °C) to avoid internalisation of CI-MPRs. A total of 10^5^ cells were washed twice with cold PBS + 1% BSA and resuspended in blocking buffer (PBS + 1% BSA with goat serum) for 30 min on ice. After two washes, cells were stained with anti IGF-II/IGF2R mouse antibody (2G11) (#NB300-514, Novus biologicals, Centennial, CO, USA) for 1.5 h on ice at 4 °C. Mouse IgG2a kappa was used as an isotype control. Cells were washed twice and incubated with a secondary antibody goat anti-mouse AlexaFluor-488 (#4408S, Cell Signaling Technology, Danvers, MA, USA) for 1 h on ice (dark). Cells were washed two times and analysed on a FACS Canto II flow cytometer (Becton Dickinson, Franklin Lakes, NJ, USA) using the FACS Diva software (BD Biosciences, Bedford, MA, USA) and FlowJo program (BD, Ashland, OR, USA).

### 4.23. Karyotypes Analysis

Karyotypes were performed by the Biobank of the Andalusian Public Health System (Granada, Spain). Cells were incubated in growth medium supplemented with 0.1 mg/mL of colcemid (Merck) for 4 h. The cytoplasm was removed using a hypotonic solution of KCl (0.075 mol/L) and the nuclei were fixed with methanol:acetic in a 3:1 ratio (vol/vol). Metaphases were fixed in slides. G-bands were made with Trypsin–Wright (GTG) and a minimum of 20 metaphases were analysed for each cell line, assigning a karyotype formula according to the international System of Human Cytogenetic Nomenclature (ISCN) 2020. Karyotypes were analysed using a Leica DM5500 microscope (Leica, Wetzlar, Germany) and the Ikaros Karyotyping System (Metasystems, Heidelberg, Germany).

### 4.24. Statistical Analysis

All the data are represented as means ± SEM. Statistical analysis was performed using GraphPad Prism software (GraphPad Software, LaJolla, CA, USA; https://www.graphpad.com, accessed on 30 April 2022). Statistical comparisons were performed by unpaired t-test (one or two tails, * *p* < 0.05, ** *p* < 0.01, *** *p* < 0.001, **** *p* < 0.0001) or ANOVA (multiple comparisons). The analysis is indicated in each figure.

## Figures and Tables

**Figure 1 ijms-23-06298-f001:**
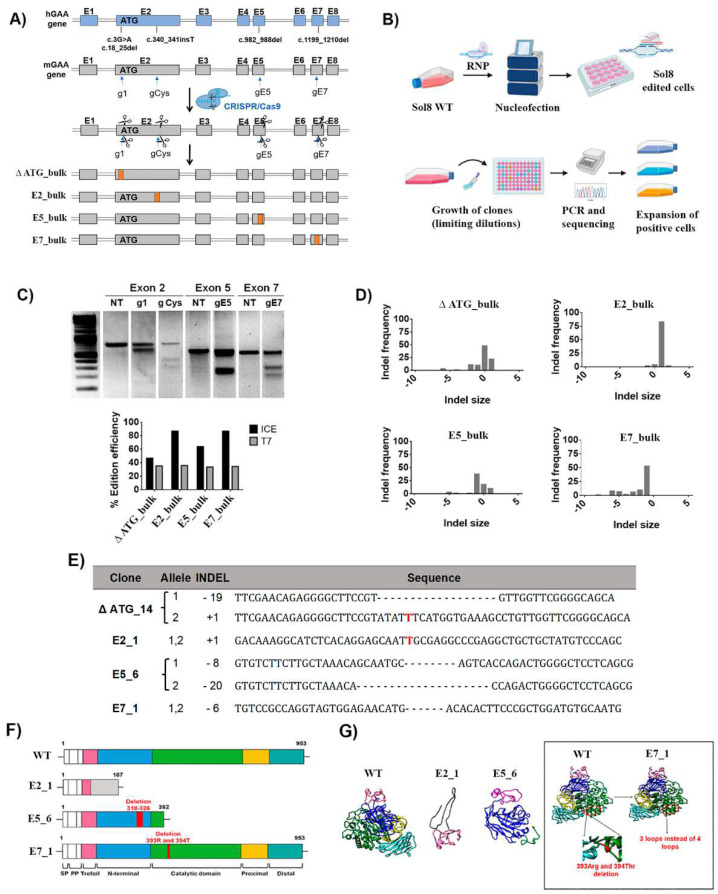
Generation of isogenic murine cellular models in Sol8 cells with severe Pompe mutations. (**A**) Scheme of the strategy designed to generate murine Sol8 cell Pompe models. These cellular models, ∆ ATG, E2, E5 and E7, resemble some mutations present in Pompe patients’ cells, such as c.3G>A and c.18_25del, c.340_341insT, c.982_988del and c.1199_1210del, respectively. These models were used to generate different protein modifications from no protein expression to different frame shifts generating aberrant proteins, or even deletions of critical amino acids at the catalytic site. (**B**) Representative scheme of the nucleofection process with RNP, cellular cloning and analysis of the cell clones for the selection of edited positive clones. Image created with BioRender.com accessed on 30 April 2022. (**C**) Editing efficiencies by CRISPR/Cas9 targeted at different exons in the murine GAA locus determined by the T7 assay and the ICE algorithm (see M&M). Top, T7EI assay of the Sol8 nucleofected cells (RNP g1-ATG, g Cys, gE5 and gE7) and NT (non-treated). Bottom, graph of the editing efficiencies in Sol8 cells nucleofected with RNP g1 (∆ ATG), g Cys (E2_bulk), gE5 (E5_bulk) and gE7 (E7_bulk) using the T7EI vs. ICE algorithm. (**D**) Graphs showing the profile of indels generated in Sol8 cells nucleofected (∆ ATG, E2_bulk, E5_bulk and E7_bulk), using the ICE algorithm. The coordinate zero represents the cut site, negative values represent deletions of different length and positive values represent insertions. (**E**) cDNA sequence in each Sol8 clone with different insertions and deletions compared with the WT Sol8 cells by ICE tool. The insertions are marked in red and the deletions as “–“. (**F**) Schematic representation of the different domains of mGAA WT, E2_1, E5_6 and E7_1 based on their homology with hGAA. (**G**) Murine GAA (WT, E2_1, E5_6 and E7_1) 3D structures predicted using SwissProt modelling. For 3D structure prediction of the different clones, we performed a translation of their sequenced mRNA using ExpasY and the structures were predicted with SwissProt modelling.

**Figure 2 ijms-23-06298-f002:**
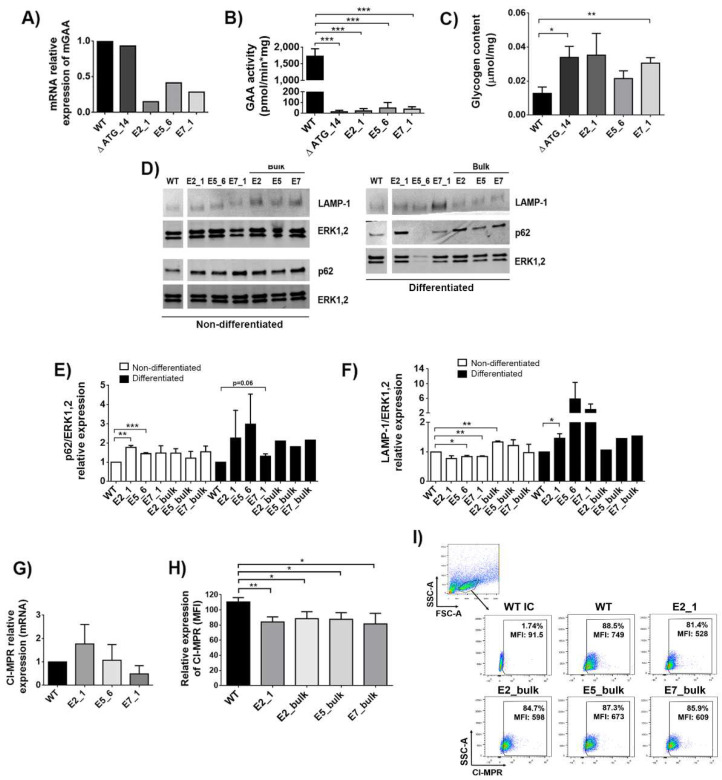
Characterisation of GAA-KO Sol8 cellular models. (**A**) mRNA relative expression of mGAA in the different Sol8 clones (∆ ATG_14, E2_1, E5_6 and E7_1) compared to WT cells. (**B**) GAA activity in Sol8 WT cells and mutated clones (∆ ATG, E2_1, E5_6 and E7_1). (**C**) Glycogen content in Sol8 myoblasts differentiated into myotubes (see M&M) in WT and clones (∆ ATG, E2_1, E5_6 and E7_1). (**D**) Western blot analysis of Sol8 cells lysates using anti-p62/sequestosome-1 (SQSTM1) and anti-LAMP-1 antibodies. Anti-ERK1,2 was used as a loading control. (**E**) Quantification of p62 levels normalised to ERK1,2. (**F**) Quantification of LAMP-1 levels normalised to ERK1,2. (**G**) mRNA relative expression of CI-MPRs in clones (∆ ATG, E2_1, E5_6 and E7_1) and Sol8 WT cells. (**H**) Surface CI-MPRs relative expression (mean fluorescence of intensity-MFI) in Sol8 WT compared to E2_1, E2_bulk, E5_bulk and E7_bulk. (**I**) Representative dot-plot of CI-MPRs expression (MFI) in Sol8 WT, E2_1, E2_bulk, E5_bulk and E7_bulk. Statistical analyses: unpaired *t*-test, two-tails (**A**–**C**), unpaired t-test one-tail (**E**–**H**) (* *p* < 0.05, ** *p* < 0.01, *** *p* < 0.001. Values represent means +/− SEM of at least 3 separate experiments.

**Figure 3 ijms-23-06298-f003:**
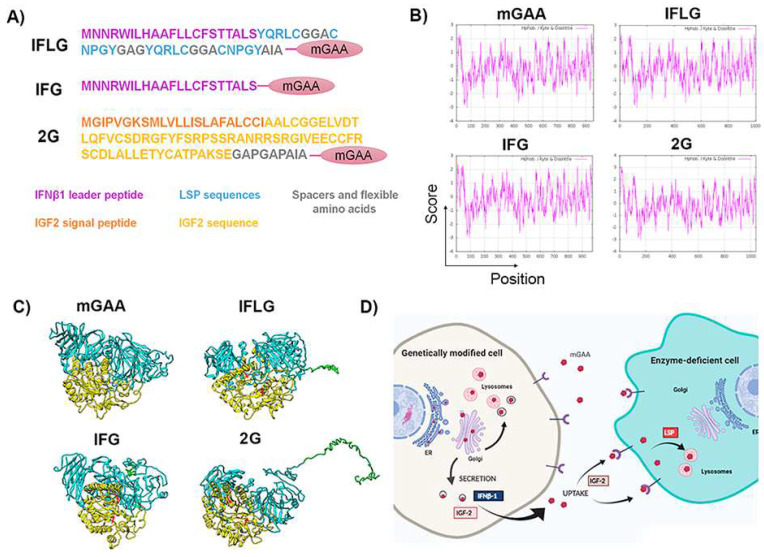
Design of different chimeric mGAA proteins to increase their secretion and lysosomal uptake. (**A**) Designs of different optimised chimeric mGAA. IFLG corresponds to the leader peptide of IFNβ1 and LSP coupled to mGAA. IFG refers to the leader peptide of IFNβ1 coupled to mGAA, and 2G corresponds to IGF2 coupled to mGAA. (**B**) Hydrophobicity pattern of mGAA, IFLG, IFG and 2G using the Kyte–Doolittle scale. The plots show hydrophilic and hydrophobic regions of native mGAA, IFLG, IFG and 2G. Values above 0 correspond to hydrophobic regions, and those below 0 correspond to hydrophilic ones. The addition of IFLG and 2G to mGAA creates a more hydrophilic region in the N-terminus than the native one. IFG has a similar tendency to the native one, due to the hydrophilicity of the leader peptide. (**C**) Three-dimensional structure prediction of optimised chimeric mGAAs by I-TASSER. The optimisations included in mGAA are present in the amino terminus of the global structure (coloured in green), coupled with the trefoil domain of mGAA, non-interfering with the catalytic domain (catalytic domain in yellow, red arrows pointing the two catalytic asparagine). (**D**) Scheme showing the rationale of GAA modifications to improve GT strategies. Created with BioRender.com. Contrary to endogenous GAA, GT-optimised GAAs (opGAA) must be primarily secreted in order to cross-correct as many cells as possible. The leader peptide of the IFNβ-1 or IGF-2 will enhance secretion of the different chimeric GAAs. Once opGAA is secreted to the medium, it must be taken up by enzyme-deficient cells to correct GAA deficiency and degrade the accumulated glycogen (cross-correction). IGF-2 and LSP sequences are included to improve uptake and/or lysosomal targeting of secreted opGAAs.

**Figure 4 ijms-23-06298-f004:**
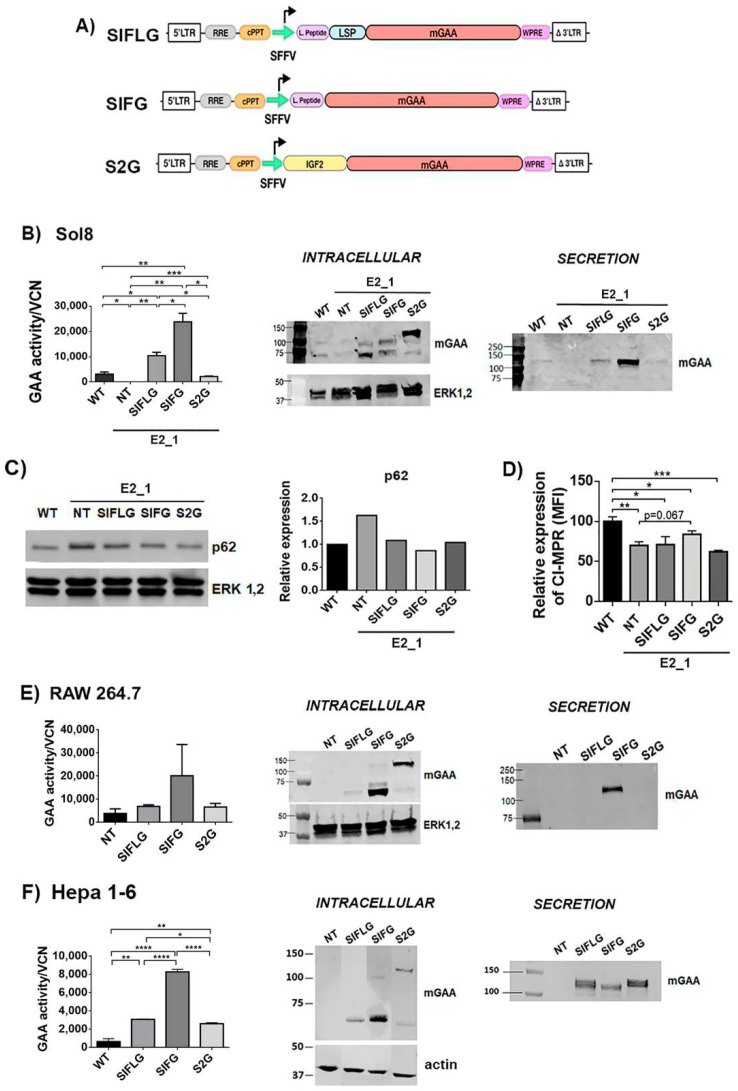
Optimisations of mGAA restore the GAA enzymatic deficit in GAA-KO cells, and it is secreted into the medium. (**A**) Schematic representation of SIFLG, SIFG and S2G constructs. mGAA, murine glucosidase alpha acid; SFFV, spleen focus forming virus promoter; WPRE, woodchuck hepatitis virus post transcriptional regulatory element. (**B**) Analysis of mGAA intracellular activity (left graph) in cell lysates (WT and NT) or after correcting by vector copy number (VCN) (SIFLG, SIFG and S2G). Middle panel and right panels: Western blots of intracellular (middle) or secreted (right) mGAA from Sol8 WT, E2_1 (NT) and E2_1 cells transduced with SIFLG, SIFG and S2G. (**C**) Western blot and quantification of p62 levels relative to ERK1,2 levels in WT, non-transduced E2_1 cells and transduced E2_1 cells with LVs. (**D**) CI-MPRs’ relative expression in rescued Sol8 GAA-KO cells (E2_1). E-F) Analysis of mGAA intracellular activity (left graph) in cell lysates (NT = WT non-transduced) or after correcting by vector copy number (VCN) (SIFLG, SIFG and S2G). Western blot of intracellular mGAA (middle panels) and secreted mGAA (right panels) of (**E**) RAW 264.7 cells and (**F**) Hepa 1-6 cells. Statistical analyses: **B**, **E** and **F**) unpaired *t*-test (two tails, * *p* < 0.05, ** *p* < 0.01, *** *p* < 0.001, **** *p* < 0.0001)). Values represent means +/− SEM of at least three separate experiments.

**Figure 5 ijms-23-06298-f005:**
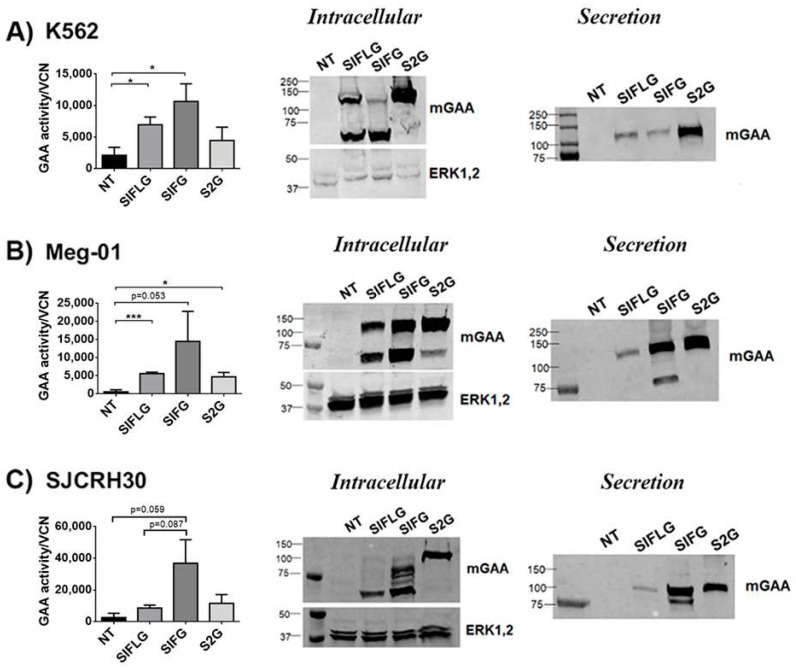
The optimised mGAA are expressed and secreted in different human cell lines. (**A**–**C**) Left graph, analysis of mGAA intracellular activity in cell lysates (NT = WT non-transduced) or after correcting by vector copy number (VCN) (SIFLG, SIFG and S2G) of transduced K562 (**A**), Meg-01 (**B**) and SJCRH30 (**C**) cells. Middle panel and right panels: Western blots of intracellular (middle) or secreted (right) mGAA from K562 (**A**) Meg-01 (**B**) and SJCRH30 (**C**) non-transduced (NT) or transduced with SIFLG, SIFG and S2G LVs, as indicated. Statistical analyses: unpaired *t*-test (two tails, **p* < 0.05, *** *p* < 0.001). Values represent means +/− SEM of at least three separate experiments.

**Figure 6 ijms-23-06298-f006:**
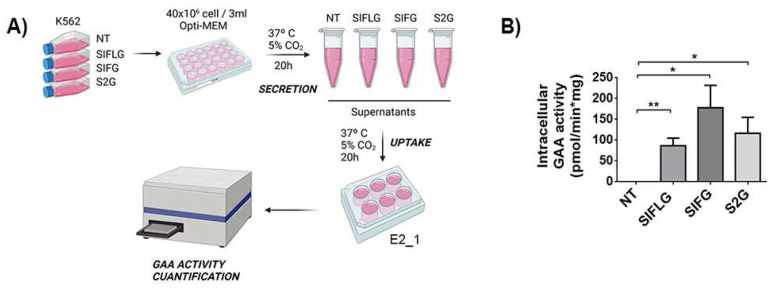
IFG-derived peptide allows a better GAA activity recovery in GAA-KO Sol8 cells. (**A**) Scheme protocol of mGAA secretion into the medium (supernatants/conditioned media) by transduced K562 cells and its uptake by E2_1 cells. Image created with BioRender.com. (**B**) mGAA intracellular activity after the addition of conditioned media to E2_1 cells. NT as a negative control (supernatant from non-transduced K562 cells). Statistical analyses: unpaired *t*-test (two tails, * *p* < 0.05, ** *p* < 0.01). Values represent means +/− SEM of at least three separate experiments.

**Figure 7 ijms-23-06298-f007:**
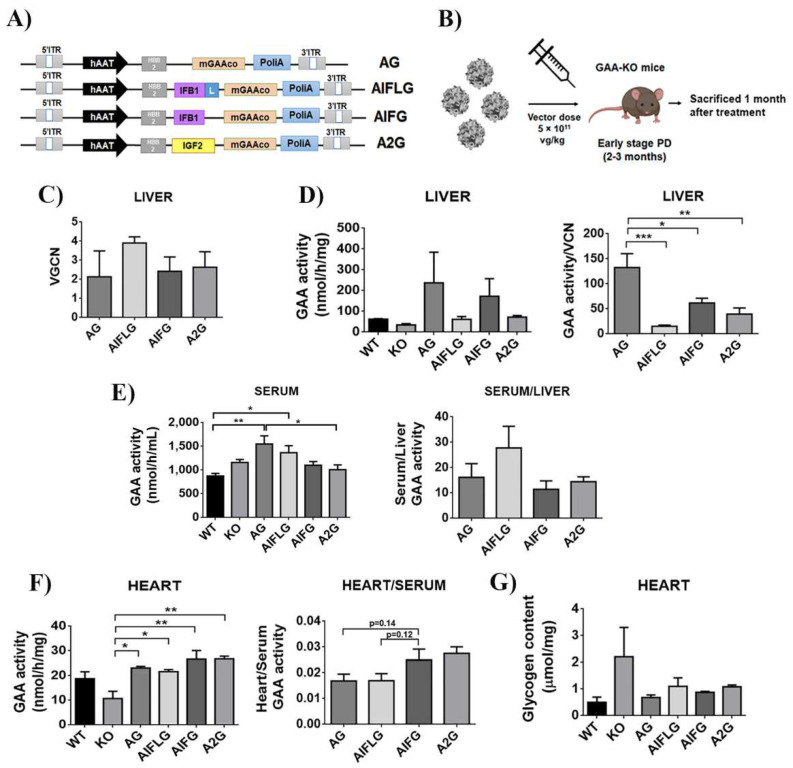
Comparison of the different GAA chimeras in GAA-KO mice. (**A**) Schematic representation of AG, AIFLG, AIFG and A2G AAV constructs. mGAA, murine glucosidase alpha acid; hAAT, human α1-antitrypsin promoter. (**B**) Workflow of in vivo experiment with AAV mGAA chimeras in GAA-KO mice. (**C**) Vector genome copy numbers (VGCN) found in the liver. (**D**) mGAA activity in the liver (left graph) and relative mGAA activity relativised by vector copy number (VGCN) in the liver (right graph). (**E**) mGAA activity in the serum (left graph) and relative levels of serum versus liver mGAA activity (right graph). (**F**) mGAA activity in the heart (left graph) and relative levels of heart versus serum mGAA activity (right graph). (**G**) Glycogen content in the heart. Statistical analyses: one-way ANOVA Tukey’s multiple comparison (* *p* < 0.05, ** *p* < 0.01, *** *p* < 0.001). Values represent means +/− SEM of at least 4 treated mice of each group.

## Data Availability

Not applicable.

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
