# Peer review of "Isogenic GAA-KO Murine Muscle Cell Lines Mimicking Severe Pompe Mutations as Preclinical Models for the Screening of Potential Gene Therapy Strategies"

_ijms, 2022, doi:10.3390/ijms23116298_

Round 1
Reviewer 1 Report
Aguilar-Gonzalez et al
General comment.
In this work the authors describe the development of murine GAA KO cell lines in which mutations found in Pompe disease patients have been introduced through gene editing. They show that these cells recapitulate some of the main biochemical and pathological features of PD (GAA deficiency, glycogen storage, secondary abnormalities of the autophagic pathway). The authors also describe the generation of GAA chimeric proteins that are tested in these cells and in vivo for their ability to correct GAA deficiency and secondary pathology in Pompe disease.
The experiments shown in this work appear in general sufficient to document and support the conclusions. The methodology used is appropriate and based on advanced approaches.
The writing of the manuscript is poor. Even a non-native English reader (like myself) recognizes frequent misspellings, typos, incorrect grammar throughout the manuscript, including abstract, figures (for example, figure 2: differenciated), and materials and methods. The manuscript would surely benefit from an extensive and accurate revision by a native English reviewer.
Specific points.
Abstract: I would suggest following the flow of the results, as they are presented in the manuscript. Specifically the generation of mutant cells should be introduced first, and followed by the generation of chimeric proteins and their effect in cells and in vivo.
Introduction: The introduction is too long. Some concepts are obvious and do not necessarily need to be mentioned (for example, the advantages of using cellular models for the study of genetic diseases). It would be advisable to shorten the introduction and to mention only the concepts that are related to the scope of the study.
Results:
The authors introduce the most severe mutations into Sol8 cells. How they define the “most severe mutations”? What are the criteria? Please specify in the text, not only in the figure legend. Are there also clinical criteria? They should be mentioned (as they are reported at the Erasmus medical center database, for example: associated with the early-onset classical infantile Pompe disease, etc).
Figure 2
Panel C (glycogen): the authors state that “all selected clones accumulated glycogen upon myotube differentiation (Figure 2C and S3B) although clone E5_6 did not reach statistical significance”. This statement is not clear. According to the figure, only deltaATG1_4 and E7_1 are significant.
Panel D (right): the quality of the western blot analysis should be improved. Apparently, there was a problem in the loading of lane E5_6. It would be advisable to provide a better quality western blot.
Panels E, F (quantitative analysis of p62, lamp-1): only in few lanes the levels of these markers are statistically different. The statement “increased p62 and LAMP-1 are indicative that autophagic build-up is occurring in our cellular models” is not fully supported by the data shown in this figure.
Line 325: “In order to analyse if potential differences in processing/secretion were tissue dependent…
It would be preferable “cell-dependent” rather than “tissue-dependent”.
First paragraph discussion: too long, does not really add useful information for the reader, redundant (some concepts have already been mentioned in the introduction). It would be better to review and shorten this section.
Discussion lines 477-492 too speculative, better to simplify.
Reviewer 2 Report
The authors aimed to a murine-murine cellular model in which to investigate murine GAA chimers prior to animal model testing of potential gene therapies for Pompe disease. The authors hope the investigation of cell models prior to testing in animal models will reduce the amount of animal experimentation needed.
The authors designed 3 different chimeric murine GAA proteins (IFG, IFLG, 2G) to be tested for therapeutic activity towards treatment of Pompe disease. The authors also generated different isogenic murine muscle cell lines that mimic the severe Pompe disease phenotype and then conducted work to validate these models.
The authors discuss some unmet needs and disadvantages of the current gold standard treatment, ERT, for Pompe disease. Unmet needs/disadvantages include lifelong cost of therapy, immunogenicity, lack of efficacy for CNS disease, and inadequate efficacy of ERT towards preventing progression of Pompe disease. The authors discuss the advantages transplantation with hematopoietic progenitor cells (HSPC) and AAV gene therapies, when compared to ERT, and how the potential improved delivery efficiency of the HSPC and AAV, when compared to ERT, might improve clinical outcomes.
The authors described authophagic build up on Pompe disease cells that ultimately impairs trafficking of CI-MPR receptors in the cell, reduces the amount of CI-MPR at the cell surface, which in turn leads to inhibition of uptake and trafficking of GAA in Pompe disease cells. Cellular markers of authophagic buildup include p62/LC3.
Importantly, the cellular models the authors were able to create displayed important aspects of the Pompe disease phenotype that may be not adequately described in some of the currently used pre-translational and translational animal models of Pompe disease. These Pompe phenotype characteristics include increased glycogen accumulation combined with increased autophagic build-up and down regulation of surface CI-MPRs.
The author’s work evaluating 3 different GAA chimeras resulted in the observation that the SIFG LVs (LV-IFG) were able to improve the representation of CI-MPR on the cell surface and normalize p62 levels (marker of authophagic buildup). They confirmed this with in vivoexperiements using liver directed AAV in GAA-KO mice. This was considered a validation of the in vitrocellular models.
The authors conclude that the isogenic murine GAA-KO models and the human cell line models that they developed hold promise as useful tools for investigating lentiviral therapies expressing GAA chimeras.. The authros conclude that the LV-IFG (SIFG) is a potential candidate for pre-translational research for gene therapies for Pompe disease.
This important work helps showcase the often unrecognized issue of authophagic build up in Pompe disease, the important role that CI-MPR receptors play, and the impact of the level of expression of CI-MPR receptors towards influencing efficacy of GAA uptake and trafficking, as well as efficacy of currently available ERT.
